# Risk Factors for the Development of Early Onset Diabetes in the Population of Sindh Province, Pakistan

**DOI:** 10.3390/biomedicines13051107

**Published:** 2025-05-02

**Authors:** Eraj Abbas, Asher Fawwad, Iftikhar Ahmed Siddiqui, Muhammad Sohail Afzal, Muhammad Ansar, Muhammad Arif Nadeem Saqib, Syed M. Shahid

**Affiliations:** 1Department of Biochemistry, Faculty of Basic Medical Sciences, Baqai Medical University, Karachi 75340, Pakistan; 2Department of Life Sciences, University of Management & Technology, Lahore 54770, Pakistan; 3Department of Biochemistry, Quaid-e-Azam University, Islamabad 45320, Pakistan; 4Department of Health Sciences Technology, National Skills University, Islamabad 44000, Pakistan; 5School of Health Science, Eastern Institute of Technology (EIT), Auckland Campus, Auckland 1010, New Zealand

**Keywords:** early-onset diabetes, risk factors, Pakistan, young adults, lifestyle, hypertension

## Abstract

**Background/Objective**: Early-onset diabetes (EOD), diagnosed at ≤35 years, is a growing public health crisis in low- and middle-income countries, including Pakistan. Identifying modifiable and non-modifiable risk factors is critical for developing effective prevention strategies. This study aimed to investigate the risk factors associated with EOD in Sindh, Pakistan, focusing on genetic, lifestyle, and metabolic determinants. **Methods**: A multicenter cross-sectional study was conducted across diabetic clinics in Sindh, with primary data collection at Baqai Institute of Diabetology and Endocrinology (Karachi, Pakistan) and secondary sites in Hyderabad, Larkana, and Sukkur. Following institutional ethical approval and informed consent, we enrolled 754 individuals (type 1 and type 2 diabetes, age at diagnosis: 15–35 years). Data on anthropometric, clinical, biochemical, and lifestyle parameters were collected via structured questionnaires. Statistical analyses included Pearson’s Chi Square tests and multivariate logistic regression in determining associations. **Results**: Logistic regression revealed key predictors of early-onset diabetes (EOD). A two-generation diabetes family history showed a strong association (aOR:1.86, 1.12–3.43). Significant lifestyle risks included physical inactivity (OR:1.40, 1.03–1.90), frequent sugary beverage intake (OR:1.93, 1.89–1.98), and abnormal sleep duration (<6 h: OR:1.58, 1.04–2.40; >8 h: OR:1.86, 1.21–2.85). Hypertension was a major metabolic predictor (elevated BP: OR:1.79, 1.28–1.54; Stage I: OR:1.81, 1.34–1.77). Cardiovascular disease and uncontrolled fasting glucose lost significance after adjustment, indicating confounding effects. **Conclusions**: This study highlights familial predisposition, sedentary behavior, poor diet, sleep disturbances, and hypertension as key contributors to EOD in young Pakistani adults. Early screening and targeted lifestyle interventions are urgently needed to mitigate this escalating epidemic.

## 1. Introduction

Diabetes mellitus (DM) is a chronic metabolic disorder characterized by persistent hyperglycemia resulting from impaired insulin secretion, insulin resistance, or both [1]. This dysregulation disrupts carbohydrate, lipid, and protein metabolism, leading to systemic complications such as microvascular damage, accelerated atherosclerosis, and end-organ dysfunction [2]. The global burden of DM has reached epidemic proportions, with an estimated 537 million cases (10.5% of adults) reported in 2021, disproportionately affecting low- and middle-income countries (LMICs), which account for 81% of cases [3,4]. Rapid urbanization, obesogenic environments, and sedentary lifestyles have fueled this rise, particularly in regions undergoing nutritional and epidemiological transitions [5].

Of particular concern is the increasing incidence of early-onset diabetes (EOD; diagnosis ≤35 years), which poses distinct clinical and public health challenges [6]. Younger patients face accelerated disease progression, higher risks of micro- and macrovascular complications (e.g., nephropathy, retinopathy, and premature cardiovascular disease), and greater lifetime healthcare demands [7,8]. The etiology of EOD is multifactorial, driven by genetic predisposition (e.g., familial aggregation, epigenetic modifications), modifiable lifestyle factors (e.g., physical inactivity, excessive sugar-sweetened beverage consumption), and developmental origins (e.g., maternal diabetes, intrauterine metabolic programming) [9,10,11].

According to the International Diabetes Federation (IDF), Pakistan ranks third globally in terms of the number of adults with diabetes, with an estimated 33 million people affected and an overall national prevalence of 26.7% among adults aged 20–79 years [12,13]. While national data indicate a growing tendency of diabetes among younger individuals in Pakistan, with nearly 18% of diagnosed cases occurring in those under the age of 40 [14], there is a significant lack of region-specific data on early-onset diabetes (EOD). In particular, no large-scale study has been conducted to investigate the onset of diabetes in the 15–35-year age group within the province of Sindh. A prior study, however, identified generalized risk factors (e.g., obesity, smoking, urbanicity); data on population-specific determinants of EOD remain scarce [15]. This gap is critical, as ethnic and regional variations in genetic susceptibility, dietary patterns, and healthcare access may modulate risk [16].

This study investigated the risk factors for EOD (ages 15–35 years) in Sindh, Pakistan, with a focus on behavioral, clinical, and sociodemographic predictors. We hypothesized that specific modifiable factors such as poor dietary habits, physical inactivity, obesity, smoking, and socioeconomic status are significantly associated with an increased risk of developing EOD in this population. By elucidating these determinants, our findings aim to inform targeted prevention strategies, mitigate long-term complications, and reduce the economic burden of DM in high-risk populations.

## 2. Materials and Methods

A multicentre, cross-sectional observational study was conducted across tertiary-care diabetic clinics in Sindh, Pakistan, with primary data collection at the Baqai Institute of Diabetology and Endocrinology (Karachi, Pakistan) and secondary sites in Hyderabad, Larkana, and Sukkur. Participants aged 15–35 years with a confirmed diagnosis of type 1 or type 2 diabetes (per ADA criteria) were enrolled. Exclusion criteria included neonatal diabetes mellitus, hypoglycemia of infancy, and maternally inherited diabetes and deafness (MIDD) to minimize confounding genetic influences.

Given the absence of prior prevalence data for this age group in Sindh, a conservative estimate of 50% prevalence was assumed. Using OpenEpi v3.0 (95% CI, 5% margin of error), the minimum sample size was n = 348. To enhance statistical power and generalizability, n = 754 participants were recruited. The study protocol was approved by the Institutional Review and Ethics Board (IREB) of Baqai Medical University (BMU-IREB/03-2023). Written informed consent was obtained from all participants prior to enrolment.

A structured questionnaire administered by trained personnel captured the sociodemographic and clinical data, including demographics (age, sex, education, occupation, household income), family history (diabetes in first- and second-degree relatives), lifestyle behaviors (physical activity per WHO guidelines), dietary habits (fruit/vegetable intake, junk food frequency), tobacco use, sleep duration, and stress levels. Anthropometric and blood pressure measurements were also recorded, like height/weight measured barefoot using a Viva Stadiometer and Beurer digital scale [17]; BMI was calculated as kg/m^2^ (adults) or percentiles (adolescents; WHO standards) [18]. Blood pressures were measured as averaged from three seated readings (5 min intervals) via a mercury sphygmomanometer, classified per ACC/AHA 2017 guidelines.

Fasting blood glucose (FBG) measured via GOD-PAP method (Selectra Pro-S analyzer, ELITech Group Inc., Puteaux, France) [19]. HbA1c quantified by HPLC (EDTA tubes) [20] by transporting the samples are 4 °C and analyzed within 24 h.

Data were analyzed using IBM SPSS v23.0 by means of descriptive statistics using frequencies, mean ± SD. Bivariate analysis was performed using Pearson’s χ^2^ tests for categorical variables. Multivariate analysis was conducted using binary logistic regression (adjusted for age and BMI) to estimate adjusted odds ratios (ORs) with 95% CIs. Statistical significance was set at *p* < 0.05. The following operational definitions were used in this study: BMI categories: underweight (<18.5 kg/m^2^), normal (18.5–24.9), overweight (25–29.9), obese (≥30) [21]; hypertension staging: per ACC/AHA criteria [22] (Normal, Elevated, Stage I/II); glycemic control: ADA-defined thresholds (HbA1c >7%; FBG 80–130 mg/dL) [23]; physical activity: WHO recommendations (≥150 min/week moderate or ≥75 min vigorous) [24]; dietary adequacy: WHO’s ≥400 g/day (5 portions) of fruits/vegetables [25].

## 3. Results

The study population (N = 754) comprised 322 cases (42.7%) of type 1 diabetes mellitus (T1DM) and 432 cases (57.3%) of type 2 diabetes mellitus (T2DM), with a mean age at diagnosis of 26.14 ± 6.53 years. Age-stratified analysis revealed the expected predominance of T1DM in younger cohorts (15–19 years) and T2DM in older participants (31–35 years; 86.7%). Notably, we observed atypical presentations, including T1DM in older age groups and T2DM in younger individuals (Figure 1).

The participant’s characteristics, categorized by age groups (15–19, 20–24, 25–29, and 30–35 years) are given in Table 1. No significant difference was seen for gender and ethnicity across the age groups, with a slightly high ratio of males (51.7%). However, a significant difference was observed in educational levels (*p* < 0.001), marital status (*p* < 0.001), and occupation (*p* < 0.001). Similarly, the family history of diabetes also showed a significant difference (*p* = 0.023). It was noted that no history of diabetes was mostly seen in the age group of 15–19 years (32.2%), while a family history of diabetes in two generations was found in the age group of 31–35 years (58.5%).

Overall, 78 (10.0%) of the participants were smokers, 83 (11.0%) were using smokeless tobacco, and 149 (19.8%) were former smokers. However, no significant difference was seen. Similarly, 501 (66.4%) participants were physically inactive, and a significant association among age groups was seen (*p* = 0.006), with inadequate physical activity being more prevalent in the older age group (30–35 years) at 71.8% compared with other age groups (Table 1).

The consumption of sugar-added beverages, fruit and vegetable intake, junk food consumption, and diabetes medications did not show statistically significant associations with age groups. 548 (72.7%) participants were having 6–8 optimal sleep. Sleep duration also showed a significant association (*p* = 0.003), with a higher percentage of participants 259 (80.2%) in the age group of 31–35 years as shown in Table 2.

The study population demonstrated significant variations in weight status across age groups (*p* < 0.001). Among participants, 29.8% (n = 225) were classified as overweight and 25.7% (n = 194) as obese. Younger age groups showed a higher prevalence of underweight (22.4%) and normal weight (49.4%) status, while older participants (31–35 years) exhibited greater proportions of overweight (34.7%) and obesity (36.4%).

Blood pressure measurements revealed significant age-related differences (*p* < 0.001). Elevated blood pressure was identified in 41.0% (n = 309) of participants, with Stage I hypertension in 19.1% (n = 144) and Stage II hypertension in 1.5% (n = 11). Notably, the highest prevalence of elevated blood pressure (47.4%) and Stage I hypertension (19.8%) occurred in the 31–35 years age group.

Metabolic abnormalities were prevalent in the study population, with hypercholesterolemia present in 48.4% (n = 365) of participants, though without significant age-related variation (*p* = 0.176). Cardiovascular disease affected 15.6% (n = 118) of the cohort and demonstrated significant association with age (*p* = 0.01), showing peak prevalence (24.8%) in the 20–25 years age group. Glycemic control analysis revealed uncontrolled fasting blood glucose levels across all age groups (*p* = 0.043). In contrast, elevated HbA1c levels (>7%) were consistently observed throughout the population without significant age-related differences (*p* = 0.357) (Table 3).

Univariate analysis identified several significant predictors of early-onset diabetes (all *p* < 0.05), including a two-generation family history of diabetes, physical inactivity, excessive sugary beverage consumption, abnormal sleep duration (<6 or >8 h/night), elevated blood pressure, stage I/II hypertension, cardiovascular disease, and uncontrolled fasting blood sugar. While the male gender showed a non-significant trend toward increased risk (*p* > 0.05), the remaining associations were statistically significant. In the multivariate model adjusted for age and BMI, only a two-generation family history of diabetes (OR: 1.96, 95% CI: 1.12–3.43) and abnormal sleep duration (OR: 1.84, 95% CI: 1.14–2.97) maintained strong independent associations. Several factors that were significant in univariate analysis—including elevated blood pressure, sugary beverage consumption, and physical inactivity—showed attenuated associations after adjustment (*p* < 0.05), suggesting potential confounding effects of age and BMI (Table 4).

## 4. Discussion

This study provides critical insights into the epidemiology of early-onset diabetes in Sindh, Pakistan, revealing a multifaceted interplay of genetic, metabolic, and lifestyle risk factors. The high prevalence of type 2 diabetes (57.3%) among young adults (15–35 years) aligns with the Second National Diabetes Survey of Pakistan, which reported the highest provincial diabetes prevalence (32.2%) in Sindh, suggesting unique regional pathophysiological mechanisms may be at play [26]. In contrast with global patterns where type 1 diabetes typically predominates under 20 years and type 2 diabetes tends to emerge after 35 years [27], our findings regarding type 1 diabetes distribution challenge conventional paradigms. While type 1 diabetes represented 42.7% of cases overall, its prevalence in adults (20–35 years) was substantial (17.3–61.2%), supporting emerging evidence of significant adult-onset type 1 diabetes. This has important diagnostic implications, as adult-onset type 1 diabetes may initially present with clinical features resembling type 2 diabetes. The absence of comprehensive epidemiological data on adult-onset type 1 diabetes in Pakistan underscores the urgent need for population-based studies incorporating autoantibody testing.

The strong association between two-generation family history and early-onset diabetes, which persisted after multivariate adjustment, reinforces the crucial role of genetic predisposition. This finding extends previous research demonstrating familial aggregation of diabetes risk and suggests that in this population, multigenerational exposure may confer greater risk than single-generation family history [28]. Notably, this association remained significant even after controlling for BMI, indicating that genetic susceptibility operates independently of adiposity in this cohort.

Physical inactivity (prevalence: 66.4%) and excessive sugary beverage consumption emerged as significant modifiable risk factors, particularly among younger participants (15–25 years). These findings highlight the need for targeted lifestyle interventions in this demographic. Abnormal sleep duration (<6 or >8 h) showed a U-shaped association with diabetes risk, supporting growing evidence [29] of circadian disruption in metabolic dysregulation.

The relationship between blood pressure abnormalities and diabetes risk revealed important nuances. While elevated blood pressure and hypertension stages I–II showed strong univariate associations, these attenuated after multivariate adjustment, suggesting that they may mediate rather than independently predict diabetes risk in this population. Cardiovascular disease demonstrated complex associations, with significant univariate but attenuated multivariate effects, aligning with recent findings on the interplay between familial cardiovascular risk and diabetes [30].

These findings have important clinical and public health implications. The high prevalence of modifiable risk factors suggests that targeted lifestyle interventions could substantially reduce diabetes incidence in this vulnerable population. Lifestyle interventions at this stage can effectively delay or prevent the progression of diabetes and its long-term complications, such as cardiovascular disease, nephropathy, and neuropathy. The emphasis on these preventable factors is essential for reducing both individual and public health burdens in the future. The significant burden of adult-onset type 1 diabetes calls for improved diagnostic protocols incorporating autoantibody testing in young adults. Family history assessment should be prioritized in clinical screening, particularly for normal-weight individuals, and integrated cardiometabolic risk reduction programs should be developed for high-risk youth.

To the best of our knowledge, this is the first study to comprehensively explore the risk factors associated with EOD among individuals aged 15–35 years across major districts of Sindh, Pakistan. No prior research from this region has specifically targeted this age group or assessed risk factors with a focus on modifiable components. Moreover, previous regional data have not clearly differentiated between types of diabetes in this age group, often leading to misclassification or underreporting. Our study explicitly distinguishes and documents confirmed cases of type 1 and type 2 diabetes based on ADA criteria and clinical follow-up, thus contributing to accurate epidemiological data and supporting the maintenance of more precise health records for young individuals with diabetes.

Several study limitations warrant consideration, including the cross-sectional design, potential recall bias in self-reported measures, lack of autoantibody testing, and single-region sampling. Nevertheless, this study provides compelling evidence that early-onset diabetes in Sindh results from complex interactions between genetic susceptibility and modifiable lifestyle factors. The findings underscore the urgent need for enhanced screening protocols, culturally tailored prevention strategies, and further research into adult-onset type 1 diabetes epidemiology to address the growing burden of early-onset diabetes in Pakistan and similar populations undergoing rapid epidemiological transition.

## 5. Conclusions and Recommendations

This study elucidates the multifactorial etiology of early-onset diabetes in Sindh, Pakistan, demonstrating significant contributions from both non-modifiable (genetic predisposition) and modifiable (lifestyle-related) risk factors. The predominance of type 2 diabetes in young adults, coupled with substantial rates of adult-onset type 1 diabetes, underscores the complex diabetes phenotype in this population. Our findings highlight the critical need for a dual approach to diabetes prevention and management in Pakistan: (1) enhanced clinical recognition of atypical presentations through improved diagnostic protocols, including family history assessment and autoantibody testing where indicated, and (2) population-level interventions targeting modifiable risk factors such as physical inactivity, unhealthy dietary patterns, and sleep hygiene. The persistent association of two-generation family history with diabetes risk, independent of BMI, suggests that genetic and epigenetic factors may play an especially important role in this population. These results call for immediate public health action, including the development of culturally appropriate, age-specific prevention programs and the integration of diabetes screening with cardiovascular risk assessment.

Future longitudinal studies incorporating objective measures of lifestyle factors and genetic markers are warranted to better understand the temporal relationships and gene-environment interactions underlying early-onset diabetes in South Asian populations. Addressing this growing epidemic will require coordinated efforts across healthcare systems, policymakers, and community organizations to implement effective prevention strategies and reduce the future burden of diabetes-related complications in Pakistan’s young population.

## Figures and Tables

**Figure 1 biomedicines-13-01107-f001:**
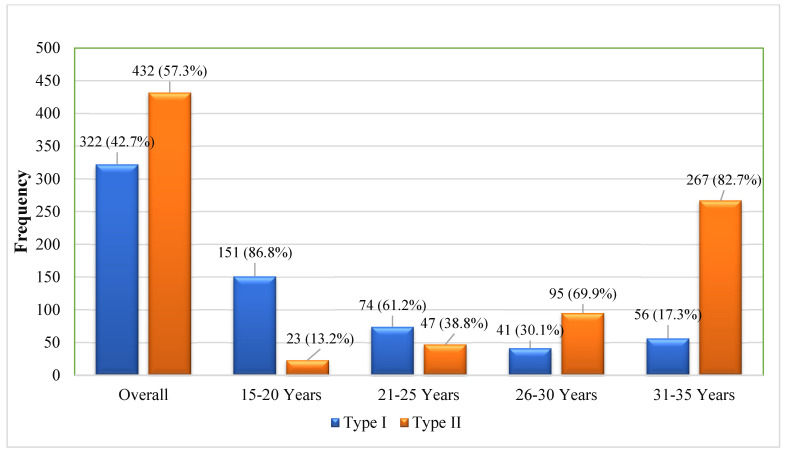
Distribution of diabetes by age group.

**Table 1 biomedicines-13-01107-t001:** Baseline characteristics of EOD (age of diagnosis <35 years).

Parameters		Overall n (%)	15–19 Years	20–24 Years	25–29 Years	30–35 Years	Chi Square/F Value	*p*-Value
Gender n (%)	Male	390 (51.7)	100 (57.5)	62 (51.2)	67 (49.3)	161 (49.8)	3.09	0.37
	Female	364 (48.3)	74 (42.5)	59 (48.8)	69 (50.7)	162 (50.2)		
Age of Diagnosis (Years)	Mean ± SD	26.14 ± 6.53	16.79 ± 1.53	22.35 ± 1.81	28.63 ± 1.46	33.20 ± 1.41	4283.17	<0.001 *
Ethnicity n (%)	Baloch	55 (7.3)	12 (6.9)	6 (5.0)	10 (7.4)	27 (8.4)	12.50	0.64
	Hindko	20 (2.7)	3 (1.7)	2 (1.7)	3 (2.2)	12 (3.7)		
	Pathan	139 18.4)	37 (21.3)	20 16.5)	28 (20.6)	54 (16.7)		
	Punjabi	103 (13.7)	24 (13.8)	20 (16.5)	19 (14.0)	40 (12.4)		
	Un Disclosed	119 (15.8)	29 (16.7)	26 (21.5)	15 (11.0)	49 (15.2)		
	Sindhi	318 (42.2)	69 (39.7)	47 38.8)	61 (44.9)	141 43.7)		
Highest Education n (%)	College/University Completed	207 (27.5)	47 (27.0)	37 (30.6)	43 (31.6)	80 24.8)	58.15	<0.001 *
	High School Completed	192 (25.5)	44 (25.3)	28 (23.1)	34 (25.0)	86 26.6)		
	No Formal Schooling	153 (20.3)	20 (11.5)	26 (21.5)	27 (19.9)	80 (24.8)		
	Post Graduate Degree	38 (5.0)	1 (0.6)	3 (2.5)	9 (6.6)	25 (7.7)		
	Primary School Completed	79 (10.5)	30 (17.2)	8 (6.6)	11 (8.1)	30 (9.3)		
	Un Disclosed	9 (1.2)	2 (1.1)	4 (3.3)	2 (1.5)	1 0.3)		
	Secondary School Completed	76 (10.1)	30 (17.2)	15 (12.4)	10 (7.4)	21 (6.5)		
Marital Status n (%)	Divorced	3 (0.4)	0 (0.0)	1 (0.8)	0 (0.0)	2 (0.6)	154.58	<0.001 *
	Married	549 (72.8)	72 (41.4)	85 (70.2)	115 (84.6)	277 (85.8)		
	Unmarried	191 (25.3)	102 58.6)	35 (28.9)	19 (14.0)	35 (10.8)		
	Widowed	11 (1.5)	0 (0.0)	0 (0.0)	2 (1.5)	9 (2.8)		
Occupation n (%)	Government Employee	14 (1.9)	1 (0.6)	3 (2.5)	3 (2.2)	7 (2.2)	102.03	<0.001 *
	Non-Government Employee	66 (8.8)	23 (13.2)	11 (9.1)	10 (7.4)	22 (6.8)		
	Self-Employed	231 (30.6)	48 (27.6)	30 (24.8)	46 (33.8)	107 (33.1)		
	Student	123 (16.3)	66 (37.9)	16 (13.2)	13 (9.6)	28 (8.7)		
	Un-Employed	320 (42.4)	36 (20.7)	61 (50.4)	64 (47.1)	159 49.2)		
District n (%)	Karachi	489 (64.9)	120 (69.0)	85 (70.2)	90 (66.2)	194 (60.1)	15.193	0.08
	Hyderabad	100 (13.3)	24 (13.8)	15 (12.4)	16 (11.8)	45 (13.9)		
	Larkana	57 (7.6)	12 (6.9)	12 (9.9)	11 (8.1)	22 (6.8)		
	Sukkar	108 (14.3)	18 (10.3)	9 (7.4)	19 (14.0)	62 (19.2)		
Family History of Diabetes n (%)	No History	217 (28.8)	56 (32.2)	36 (29.8)	35 (25.7)	90 (27.9)	9.50	0.023 *
	One Generation	62 (8.2)	13 (7.5)	8 (6.6)	13 (9.6)	28 (8.7)		
	Two Generations	421 (55.8)	86 (49.4)	67 (55.4)	79 (58.1)	189 (58.5)		
	Three Generations	54 (7.2)	19 (10.9)	10 (8.3)	9 (6.6)	16 (5.0)		

* *p* < 0.05 was considered statistically significant using the Pearson Chi Square test.

**Table 2 biomedicines-13-01107-t002:** Analysis of behavioral factors among EOD (age of diagnosis <35 years).

Parameters		Overall n (%)	Age of Diagnosis of Diabetes				Chi Square/Z	*p*-Value
			15–19 Years	20–25 Years	26–30 Years	31–35 Years		
Smoking n (%)	Yes	78 (10.0)	15 (8.6)	17 (14.0)	15 (11.0)	31 (9.6)	2.612	0.45
	No	676 (90.0)	159 (91.4)	104 (86.0)	121 (89.0)	292 (90.4)		
Smokeless Tobacco Use n (%)	Yes	83 (11.0)	21 (12.1)	16 (13.2)	15 (11.0)	31 (9.6)	1.46	0.69
	No	671 (89.0)	153 (87.9)	105 (86.8)	121 (89.0)	292 (90.4)		
Former Smokers n (%)	Yes	149 (19.8)	27 (15.5)	23 (19.0)	27 (19.9)	72 (22.3)	3.32	0.34
	No	605 (80.2)	147 (84.5)	98 (81.0)	109 (80.9)	251 (77.7)		
Physical Activity n (%)	Inadequate	501 (66.4)	117 (67.2)	66 (54.5)	86 (63.2)	232 (71.8)	12.558	0.006 *
	Adequate	232 (33.6)	57 (32.8)	55 (45.5)	50 (36.8)	91 (28.2)		
Consumption of sugar-added beverages? n (%)	Never	370 (49.1)	79 (45.4)	66 (54.5)	62 (45.6)	163 (50.5)	16.04	0.379
	Daily	62 (8.2)	14 (8.0)	10 (8.3)	7 (5.1)	31 (9.6)		
	One a Month	46 (6.1)	11 (6.3)	6 (5.0)	11 (8.1)	18 (5.6)		
	One a Week	18 (2.1)	2 1.1)	4 (3.3)	2 (1.5)	10 (3.1)		
	Rarely	9 (1.2)	0 (0.0)	2 (1.7)	2 (1.5)	5 (1.5)		
	Twice a Week	249 (33.0)	68 (39.1)	33 (27.3)	52 (38.2)	96 (29.7)		
Fruit intake n (%)	Adequate	445 (59.0)	99 (56.9)	73 (60.3)	82 (60.3)	191 (59.1)	0.50	0.91
	Inadequate	309 (41.0)	75 (43.1)	48 (39.7)	54 (39.7)	132 (40.9)		
Vegetable intake n (%)	Adequate	480 (63.7)	111 (63.8)	84 (69.4)	81 (59.6)	204 63.2)	2.76	0.43
	Inadequate	274 (36.3)	63 (36.2)	37 (30.6)	55 (40.4)	119 (36.8)		
Consumption of Junk food n (%)	Daily	165 (21.9)	37 (21.3)	28 (23.1)	25 (18.4)	75 (23.2)	14.02	0.524
	Never	123 (16.3)	37 (21.3)	17 (14.0)	20 (14.7)	49 (15.2)		
	One a Month	106 (14.1)	26 (14.9)	21 (17.4)	22 (16.2)	37 (11.5)		
	One a Week	150 (19.9)	33 (19.0)	25 (20.7)	25 (18.4)	67 (20.7)		
	Rarely	117 (15.5)	25 (14.4)	20 (16.5)	21 (15.4)	51 (15.8)		
	Twice a Week	93 (12.3)	16 (9.2)	10 (8.3)	23 (16.9)	44 13.6)		
Sleep Duration n (%)	6–8 h	548 (72.7)	118 (67.8)	74 (61.2)	97 (71.3)	259 (80.2)	20.23	0.003 *
	>8 h	106 (14.1)	29 (16.7)	22 (18.2)	20 (14.7)	35 (10.8)		
	<6 h	100 (13.3)	27 (15.5)	25 (20.7)	19 (14.0)	29 (9.0)		

* *p* < 0.05 was considered statistically significant using the Pearson Chi Square test.

**Table 3 biomedicines-13-01107-t003:** Analysis of metabolic and biochemical factors among EOD (age of diagnosis <35 years).

Parameters		Overall n (%)	Age of Diagnosis of Diabetes				Chi Square/Z	*p*-Value
			15–19 Years	20–25 Years	26–30 Years	31–35 Years		
BMI Levels, n (%)	Underweight	80 (10.6)	39 (22.4)	15 (12.4)	12 (8.8)	14 (4.3)	113.3	<0.001 *
	Normal weight	255 (33.8)	86 (49.4)	54 (44.6)	35 (25.7)	80 (24.8)		
	Overweight	225 (29.8)	37 (21.3)	29 (24.0)	47 (34.6)	112 34.7)		
	Obese	194 (25.7)	12 (6.9)	23 (19.0)	42 (30.9)	117 (36.2)		
Blood Pressure, n (%)	Normal	290 (38.5)	92 (52.9)	56 (46.3)	44 (32.4)	98 (30.3)	33.41	<0.001 *
	Elevated	309 (41.0)	53 (30.5)	43 (35.5)	60 (44.1)	153 (47.4)		
	Stage-I hypertension	144 (19.1)	28 (16.1)	21 (17.4)	31 (22.8)	64 19.8)		
	Stage-II hypertension	11 (1.5)	11 (1.5)	1 (0.6)	1 (0.8)	1 (0.7)		
Hypercholesterolemia n (%)	Yes	365 (48.4)	83 (49.4)	72 (58.0)	61 (44.9)	149 (46.2)	4.95	0.17
	No	389 (51.6)	88 (50.6)	52 (42.0)	75 (55.1)	174 53.9)		
Cardiovascular Disease, n (%)	Yes	118 (15.6)	27 (15.5)	30 (24.8)	20 (14.7)	41 (12.7)	9.90	0.01 *
	No	636 (84.3)	147 (84.5)	91 (75.2)	116 (85.3)	282 87.3)		
FBS Level n (%)	Controlled	176 (38.5)	38 (36.2)	17 (25.8)	40 (47.1)	81 (40.3)	7.66	0.04 *
	Uncontrolled	281 (61.5)	67 (63.8)	49 (74.2)	45 (52.9)	120 (59.7)		
HbA1c Levels n (%)	Controlled	65 (8.6)	10 (5.7)	9 (7.4)	13 (9.6)	33 (10.2)	3.23	0.35
	Uncontrolled	689 (91.4)	164 (94.3)	112 (92.6)	123 (90.4)	290 (89.8)		

* *p* < 0.05 was considered statistically significant using the Pearson Chi Square test.

**Table 4 biomedicines-13-01107-t004:** Risk assessment of EOD with studied factors.

Risk Factors	Univariate Odds Ratio (95% CI)	Multivariate ^£^ Odds Ratio (95% CI)
Male	1.11 (0.83–1.48)	0.82 (0.59–1.16)
Two Generations with Diabetes	1.86 * (1.15–3.02)	1.96 * (1.12–3.43)
In Adequate Physical Activity	1.40 * (1.03–1.90)	0.69 * (0.49–0.98)
Sugary Beverages	1.93 * (1.89–1.98)	1.94 * (1.89–1.99)
Sleep Duration > 8 h	1.58 * (1.04–2.40)	1.68 * (1.04–2.72)
Sleep Duration < 6 h	1.86 * (1.21–2.85)	1.84 * (1.14–2.97)
Elevated BP	1.79 * (1.28–1.54)	1.64 * (1.44–1.94)
Stage-I Hypertension	1.81 * (1.34–1.77)	1.70 (1.11–1.44)
Stage-II Hypertension	1.78 * (1.03–1.85)	1.64 (1.09–2.28)
Cardiovascular Disease	1.88 * (1.12–3.16)	1.38 (0.75–2.50)
Uncontrolled FBS	1.52 * (1.03–2.25)	0.83 (0.53–1.31)

Dependent variable: Early onset diabetes with age 15–26 years old. * Odds ratio was considered statistically significant with *p* < 0.05. £: Multivariate model was adjusted for age and BMI.

## Data Availability

The original contributions presented in this study are included in the article. Further inquiries can be directed at the corresponding author.

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
