# Peer review of "Risk Factors for the Development of Early Onset Diabetes in the Population of Sindh Province, Pakistan"

_biomedicines, 2025, doi:10.3390/biomedicines13051107_

Round 1

Reviewer 1 Report

Comments and Suggestions for Authors

This article provides a comprehensive and valuable analysis of the epidemiology of early-onset diabetes in Sindh, Pakistan, effectively identifying and examining a range of genetic, metabolic, and lifestyle-related risk factors. The high prevalence of type 2 diabetes among young individuals, combined with modifiable factors such as excessive sugary beverage consumption, physical inactivity, and sleep disturbances, paints a concerning picture of a growing health crisis. By highlighting these risk factors, the study not only raises awareness but also frames diabetes as a pressing public health issue in the region—one that demands immediate interventions, targeted policies, and widespread screening efforts.

  • The results of this study show that there was no significant difference based on gender, please explain how the gender imbalance (57.5% male) may have introduced a bias?

  • The results also reported that dietary habits (consumption of sugary drinks, fruits, vegetables, fast food) were not significantly associated with age. However, a more detailed analysis (e.g., total calorie intake, daily carbohydrate intake, etc.) may be appropriate.

  • Also, high cholesterol levels were common but did not differ significantly with age. Could this be due to drug treatment? Or are genetic factors dominant?

  • An interesting point in the article is the relatively high prevalence of type 1 diabetes in adults. However, no details are provided on how to differentiate these two types of diabetes, either clinically or through testing. The data appears to be based on self-reports or initial diagnoses, which may lead to misclassification. Please elaborate.

Author Response

Comment 1: The results of this study show that there was no significant difference based on gender, please explain how the gender imbalance (57.5% male) may have introduced a bias?

Response: We would like to clarify that the gender distribution in our study population was 51.7% male and 48.3% female. This slight difference was not statistically significant, indicating a relatively balanced representation of both genders. Therefore, it is unlikely that gender distribution introduced any meaningful bias in the study outcomes.

Comment 2: The results also reported that dietary habits (consumption of sugary drinks, fruits, vegetables, fast food) were not significantly associated with age. However, a more detailed analysis (e.g., total calorie intake, daily carbohydrate intake, etc.) may be appropriate.

Response: In the current study, dietary habits were assessed using a simplified food frequency approach based on the guidelines provided by the World Health Organization (WHO), focusing on the frequency of consumption of specific food items such as sugary drinks, fruits, vegetables, and fast food. While no significant association with age was observed based on this approach, we acknowledge that this method may not capture the full nutritional profile of participants.

A more detailed dietary assessment, such as calculating total daily caloric intake, macronutrient distribution (particularly carbohydrate intake), and glycemic load, would indeed provide a deeper understanding of dietary patterns and their relationship with age and diabetes risk. We plan to incorporate such detailed nutritional analyses in future studies.

Comment 3: Also, high cholesterol levels were common but did not differ significantly with age. Could this be due to drug treatment? Or are genetic factors dominant?

Response: The widespread presence of elevated cholesterol levels regardless of age may point to a strong genetic predisposition, such as familial hypercholesterolemia or other inherited lipid abnormalities.

Comment 4: An interesting point in the article is the relatively high prevalence of type 1 diabetes in adults. However, no details are provided on how to differentiate these two types of diabetes, either clinically or through testing. The data appears to be based on self-reports or initial diagnoses, which may lead to misclassification. Please elaborate.

Response: 

We would like to clarify that all participants included in the study were confirmed cases of either Type 1 or Type 2 diabetes, diagnosed according to the American Diabetes Association (ADA) criteria. These diagnoses were not based on self-reports but were verified through medical records and regular clinical follow-up at tertiary care hospitals. The classification was made by treating endocrinologists based on comprehensive clinical evaluation and treatment response over time. As such, the likelihood of misclassification is minimal, and it was also mentioned in the manuscript on Page 2, line 30.

Reviewer 2 Report

Comments and Suggestions for Authors

The authors describe their work to investigate the risk factors associated with early-onset diabetes (EOD) in Sindh, Pakistan, focusing on genetic, lifestyle, and metabolic determinants. In this multicentre cross-sectional study involving 754 individuals (Type 1 and Type 2 diabetes, age at diagnosis: 15–35 years) it was found that a two-generation diabetes family history showed strong association. Significant lifestyle risks included physical inactivity, frequent sugary beverage intake and abnormal sleep duration. Hypertension was a major metabolic predictor. Cardiovascular disease and uncontrolled fasting glucose lost significance after adjustment, indicating confounding effects. It was concluded that familial predisposition, sedentary behaviour, poor diet, sleep disturbances, and hypertension as key contributors to EOD in young Pakistani adults. Furthermore, early screening and targeted lifestyle interventions was suggested to be urgently needed to mitigate this escalating epidemic. This is an interesting study. Appropriate methodology has been employed and the conclusions appear to be justified based on the data at hand. However, I have a few recommendations for consideration.

  1. Introduction. Can the authors please provide a clear hypothesis to be tested in the study?
  2. Introduction. To put the study into a better perspective, can the authors also provide some local statistics on the prevalence of EOD, as well as incidence/prevalence of diabetes in the population (appears to be around 13%) and how this relates to global statistics?
  3. Results/Discussion. Can the authors explore the influence of sex and ethnicity on EOD?
  4. Discussion. The authors should elaborate on the potential mechanisms that may be involved and if they are related to sex?
  5. Discussion. The authors need to emphasize and elaborate on the novelty of their study.
  6. Discussion. Please expand on the clinical and public health applicability of the findings.
  7. Discussion. Since EOD is now a global phenomena, the authors should describe and compare their regional findings with other jurisdictions.

Author Response

Comment 1: 

  1. The authors describe their work to investigate the risk factors associated with early-onset diabetes (EOD) in Sindh, Pakistan, focusing on genetic, lifestyle, and metabolic determinants. In this multicentre cross-sectional study involving 754 individuals (Type 1 and Type 2 diabetes, age at diagnosis: 15–35 years) it was found that a two-generation diabetes family history showed strong association. Significant lifestyle risks included physical inactivity, frequent sugary beverage intake and abnormal sleep duration. Hypertension was a major metabolic predictor. Cardiovascular disease and uncontrolled fasting glucose lost significance after adjustment, indicating confounding effects. It was concluded that familial predisposition, sedentary behaviour, poor diet, sleep disturbances, and hypertension as key contributors to EOD in young Pakistani adults. Furthermore, early screening and targeted lifestyle interventions was suggested to be urgently needed to mitigate this escalating epidemic. This is an interesting study. Appropriate methodology has been employed and the conclusions appear to be justified based on the data at hand. However, I have a few recommendations for consideration.

Introduction. Can the authors please provide a clear hypothesis to be tested in the study?

Response: The clear hypothesis has been incorporated in the introduction section of the revised manuscript at Page 2, Line 27.

Comment 2: Introduction. To put the study into a better perspective, can the authors also provide some local statistics on the prevalence of EOD, as well as incidence/prevalence of diabetes in the population (appears to be around 13%) and how this relates to global statistics?

Response: The response has been incorporated in the introduction section of the revised manuscript at Page 2, Line 15.

Comment 3: Results/Discussion. Can the authors explore the influence of sex and ethnicity on EOD?

Response: We have already explored the influence of sex and ethnicity on EOD in our analysis. The statistical findings indicated no significant association between either sex or ethnicity and the risk of EOD (p > 0.05). These results are reflected in the results and discussed in the revised manuscript.

Comment 4: Discussion. The authors should elaborate on the potential mechanisms that may be involved and if they are related to sex?

Response: The discussion section of the revised manuscript has been modified with suggested changes.

Comment 5: Discussion. The authors need to emphasize and elaborate on the novelty of their study.

Response: The response is added to the discussion. For the clarity of the novelty of this study, we would like to mention that, to the best of our knowledge, this is the first study to comprehensively explore the risk factors associated with EOD among individuals aged 15–35 years across major districts of Sindh, Pakistan. No prior research from this region has specifically targeted this age group or assessed risk factors with a focus on modifiable components. This age range is particularly important, as it presents a critical window of opportunity for intervention. Many of the identified risk factors, such as unhealthy diet, physical inactivity, and sleep duration, are preventable and modifiable. Lifestyle interventions at this stage can effectively delay or prevent the progression of diabetes and its long-term complications, such as cardiovascular disease, nephropathy, and neuropathy. The emphasis on these preventable factors is essential for reducing both individual and public health burdens in the future. Moreover, previous regional data have not differentiated between types of diabetes in this age group, often leading to misclassification or underreporting. Our study explicitly distinguishes and documents confirmed cases of Type 1 and Type 2 diabetes based on ADA criteria and clinical follow-up, thus contributing to accurate epidemiological data and supporting the maintenance of more precise health records for young individuals with diabetes.

Comment 6: Discussion. Please expand on the clinical and public health applicability of the findings.

Response: The response has been incorporated in the discussion section of the revised manuscript

Comment 7: Discussion. Since EOD is now a global phenomenon, the authors should describe and compare their regional findings with other jurisdictions.

Response: This point has been addressed and incorporated in the discussion section of revised manuscript.

Round 2

Reviewer 1 Report

Comments and Suggestions for Authors

-

Reviewer 2 Report

Comments and Suggestions for Authors

The authors have addressed all concerns and have adequately revised their manuscript. I have no further comments.